**Data Availability Statement:** All relevant data are within the manuscript.

**Funding:** This project was supported financially by the Academy of Scientific Research and Technology (ASRT), Egypt, under initiatives of

# Enhanced quantum signature scheme using quantum amplitude amplification operators

**Basma Elias**[1,2☯]*, **Ahmed Younes**[2,3,4☯]

**1** Department of Mathematics, Faculty of Education, Alexandria University, Alexandria, Egypt, **2** Academy of Scientific Research and Technology(ASRT), Cairo, Egypt, **3** Department of Mathematics and Computer Science, Faculty of Science, Alexandria University, Alexandria, Egypt, **4** School of Computer Science, University of Birmingham, Birmingham, United Kingdom

☯ These authors contributed equally to this work.
\* basmaaelias@alexu.edu.eg

## Abstract

Quantum signature is the use of the principles of quantum computing to establish a trusted communication between two parties. In this paper, a quantum signature scheme using amplitude amplification techniques will be proposed. To secure the signature, the proposed scheme uses a partial diffusion operator and a diffusion operator to hide/unhide certain quantum states during communication. The proposed scheme consists of three phases, preparation phase, signature phase and verification phase. To confuse the eavesdropper, the quantum states representing the signature might be hidden, not hidden or encoded in Bell states. It will be shown that the proposed scheme is more secure against eavesdropping when compared with relevant quantum signature schemes.

## 1 Introduction

Digital signature is an important branch of cryptography [1, 2]. Classical signature is used in applications such as e-mails and E-payment systems [1]. The security is decreased by the development of supercomputers and/or quantum computing such as Shor's algorithm for factoring integers [3–5]. Signature schemes are classified based on whether they use a known message or an unknown message [6]. Quantum signature is an alternative to digital signature and takes advantages of quantum mechanics to provide unconditionally secure information exchange [7].

Many advances have been made in quantum cryptography in recent years. In 2001, Terhal et al. proposed a scheme for hiding bits in Bell states [8]. In 2002, Zeng et al. proposed arbitrated quantum signature scheme (AQS). The scheme is implemented by a symmetric quantum key cryptosystem and Greenberger-Horne-Zenilinger (GHZ), the scheme could be used to sign both known and unknown quantum states [9]. In 2003, Lee et al. proposed two quantum signature schemes, one scheme uses a public board to recover the message, and the other scheme does not use a public board to recover the message, where Bob recovers the message after the arbitrator's verification [10]. In 2009, Li et al. presented another AQS scheme based on Bell states, the scheme replaces GHZ states with Bell ones [11]. In 2011, Younes proposed a

Science Up Faculty of Science (Grant No 6563). (ASRT) is the 2nd affiliation of this research. The funders had a role in the decision to publish the manuscript.

**Competing interests:** The authors have declared that no competing interests exist.

method to hide specific quantum states in a superposition using a single iteration of the amplitude amplification operators [12]. In 2014, Yoon uses Grover's quantum search algorithm to propose a quantum signature scheme [13, 14].

The aim of this paper is to use the method proposed in [12] to propose a new quantum signature scheme that depends on representing a message using a quantum state. The message can be either hidden or not hidden from a superposition during data transmission using amplitude amplification operators. The message can also be represented in Bell states. The scheme depends on a key that is shared using quantum key distribution (QKD) [15, 16] between the communication parties. The proposed quantum signature scheme has a high probability of detecting any eavesdropping.

The paper is organized as follows: Section 2 briefs the method proposed in [12] to secure transferred data through hiding quantum states, and reviews the process of hiding quantum states and showing the hidden states. Section 3 proposes the quantum signature scheme based on the diffusion, the partial diffusion operators and Bell state. Section 4 gives an analysis of the proposed signature scheme and compares the results with other quantum signatures schemes in literature. Section 6 concludes the paper.

## 2 Secure quantum communications channels

In 2011, A.Younes [12] proposed an algorithm to secure encrypted message during data transmission between two parties without using EPR source by hiding/unhiding the encrypted message. After two parties encrypt the message using any encryption algorithm, they exchange the message one bit at a time using two noiseless quantum channels. The sender chooses whether the encrypted message will be hidden or send it without any action. The hiding/ unhiding process depends on operators that are used in quantum searching algorithms. Partial diffusion operator $D_p$ [17] is used to hide a specific quantum state and diffusion operator $G$ [14] is used to unhide the hidden quantum states. Further, The processes of hiding and unhiding will be explained later.

To send a bit using the proposed algorithm, Alice prepares the system $D_p U_f |00\rangle \otimes |0\rangle$. Alice chooses $U_f$ according to three criteria: the data to be sent is 0 or 1, the data is on the first or second qubit, and whether the quantum state represents the data will be hidden or not. Alice and Bob share a secret key of the form $(A, p)$ where $A$ is an action that represents hide the message or not hide it, actions denoted as $H$ and $N$ respectively, and $p$ is the bit position, $p$ can take one of two values either 1 for the first bit or 2 for the second bit. For example, if the key is $(N, 1)$, this means Alice sends the data on the first qubit and the data is not hidden, then Bob directly reads the data from the first qubit. If the key is $(H, 2)$, it means Alice sends the data on the second qubit and the data is hidden, then Bob applies $G$ then reads the data from the second qubit.

### 2.1 Hiding quantum state

Given an $n$ qubit quantum system in a superposition. It is required to hide certain quantum states from that superposition. To achieve the aim, an extra qubit initialized to state $|0\rangle$ is appended to the system, where the extra qubit will be used to mark the required quantum states via entanglement, then apply $D_p$ operator. Assume an operator $U_f$ that evaluates to 1 for the required states, where

$$U_f|x, 0\rangle = \begin{cases} |x, 1\rangle & , f(x) = 1, \\ |x, 0\rangle & , f(x) = 0. \end{cases} \tag{1}$$

The $D_p$ operator performs inversion about the mean on the states entangled with an extra qubit in state $|0\rangle$ and phase shift of -1 on the states with extra qubit $|1\rangle$. If $D_p$ is applied on a general system $|\psi\rangle$ of $n + 1$ qubits, then

$$
\begin{aligned}
D_p|\psi\rangle &= (H^{\otimes n} \otimes I1)(2|0\rangle\langle 0| - I_{n+1})(H^{\otimes n} \otimes I_1)\sum_{k=0}^{2N-1}\delta_k|k\rangle \\
&= 2(H^{\otimes n} \otimes I_1|0\rangle\langle 0|H^{\otimes n} \otimes I_1)\sum_{k=0}^{2N-1}\delta_k|k\rangle - \sum_{k=0}^{2N-1}\delta_k|k\rangle \\
&= \sum_{j=0}^{N-1}2\langle\alpha\rangle(|j\rangle \otimes |0\rangle) - \sum_{k=0}^{2N-1}\delta_k|k\rangle \\
&= \sum_{j=0}^{N-1}(2\langle\alpha\rangle - \alpha_j)(|j\rangle \otimes |0\rangle) - \sum_{j=0}^{N-1}\beta_j(|j\rangle \otimes |1\rangle),
\end{aligned}
\tag{2}
$$

where $\alpha_j = \delta_k$: $k$ even, $\beta_j = \delta_k$: $k$ odd, and $\langle\alpha\rangle = \frac{1}{N}\sum_{j=0}^{N-1}\alpha_j$ is the mean of the amplitudes of the subspace entangled with the extra qubit in state $|0\rangle$. For example, assume the following 2-qubit system,

$$
|\psi\rangle = \frac{1}{2}(|00\rangle + |01\rangle + |10\rangle + |11\rangle),
\tag{3}
$$

where it is required to hide states $|10\rangle$ and $|11\rangle$ from the superposition. Append an extra qubit initialized to state $|0\rangle$ to the system, then applies $U_f$ that evaluates to 1 for $|00\rangle$ and $|01\rangle$ then

$$
U_f|\psi\rangle = \frac{1}{2}(|10\rangle + |11\rangle) \otimes |0\rangle + \frac{1}{2}(|00\rangle + |01\rangle) \otimes |1\rangle,
\tag{4}
$$

and $\langle\alpha\rangle = \frac{1}{4}$. Applying $D_pU_f|\psi\rangle$ gives a superposition that contains only the states that should not be hidden, i.e. $D_pU_f|\psi\rangle = \frac{1}{\sqrt{2}}(|00\rangle + |01\rangle) \otimes |1\rangle$, where the third qubit can be omitted since it is not part of the working system.

## 2.2 Unhiding quantum states

To show the hidden states, we use the diffusion operator $G$ that performs the usual inversion about the mean as shown in [12]. If $G$ is applied on a general system $|\psi\rangle$ of $n$ qubit, then

$$
G|\psi\rangle = \sum_{j=0}^{N-1}[-\alpha_j + 2\langle\alpha\rangle]|j\rangle,
\tag{5}
$$

where $\langle\alpha\rangle = \frac{1}{N}\sum_{j=0}^{N-1}\alpha_j$ is the mean of the amplitudes of the states in the superposition, i.e. each amplitude $\alpha_j$ will be changed based on the relation,

$$
\alpha_j \rightarrow [-\alpha_j + 2\langle\alpha\rangle].
\tag{6}
$$

There are many cases for the system $G$ to work on. The first case is a system already exists in a superposition, for example, if $|\Psi\rangle = \frac{1}{2}(|00\rangle + |01\rangle + |10\rangle + |11\rangle)$, then applying $G$ has no effect on system.

The second case is a system in the form $|\psi\rangle = |x\rangle$ such that $x$ is one of the possible states {00,01,10,11}, then applying $G$ will create a superposition of all possible states with a phase shift of -1 on $|x\rangle$. For example, if $|\Psi\rangle = |00\rangle$, then after applying $G$ the system will be $G|\Psi\rangle = \frac{-1}{2}|00\rangle + \frac{1}{2}(|01\rangle + |10\rangle + |11\rangle)$.

The third case is the system in the form $|\psi\rangle = \frac{1}{\sqrt{2}}(|x\rangle + |y\rangle)$ where $x, y$ are any two different possible states, then $G$ will transfer the amplitudes to the states that did not exist in the superposition before applying $G$. For example, if the system $|\Psi\rangle = \frac{1}{\sqrt{2}}(|00\rangle + |01\rangle)$, then after applying $G$, any hidden states in the superposition will appear and the existing states will disappear, i.e. $\frac{1}{\sqrt{2}}(|10\rangle + |11\rangle)$.

## 3 The proposed quantum signature scheme based on amplitude amplification operators

Assume Alice and Bob want to exchange a message using the proposed signature scheme. The signature scheme consists of three phases: preparation phase, signature phase and verification phase. These three phases are operated between Alice and Trusted Center (TC), and between TC and Bob.

The proposed quantum signature scheme uses a method that was proposed in [12] to sign the message and secure the signature by confusing any eavesdropper. The method secures the communication between two parties based on hiding and unhiding quantum states by using a single iteration of the amplitude amplification operators. The proposed quantum signature scheme does not only hide or not hide quantum states, but it also encodes a signature using Bell states, this increases the security of the channel because of increasing the possibilities to sign message before sending to the receiver.

In the preparation phase, a message is announced and a secret key is shared between the sender and the receiver. Alice sends the message to the receiver on a public channel. In the signature phase, the sender signs the message using the shared secret key prepared and sent by the receiver to the sender. In the verification phase, the receiver applies an operation on the signed message according to the shared key and verifies the signature by comparing the result of the operation with the public message as shown in Fig 1.

### 3.1 Phases between Alice and TC

**3.1.1 Preparation phase.** Alice announces the message that she wants to send to Bob with size $S$ bits using a classical channel.

TC sends a secret key $K_{TA}$ to Alice using QKD protocol, the size of the key after Alice and TC apply QKD protocol is $3S$ qubits. TC chooses any QKD protocol to send the key, i.e BB84. If TC chooses to use BB84 protocol to send the key, TC needs to start with a $12S$ qubit stream to get the key with size $3S$. The key represents a sequence of actions that will be taken on the

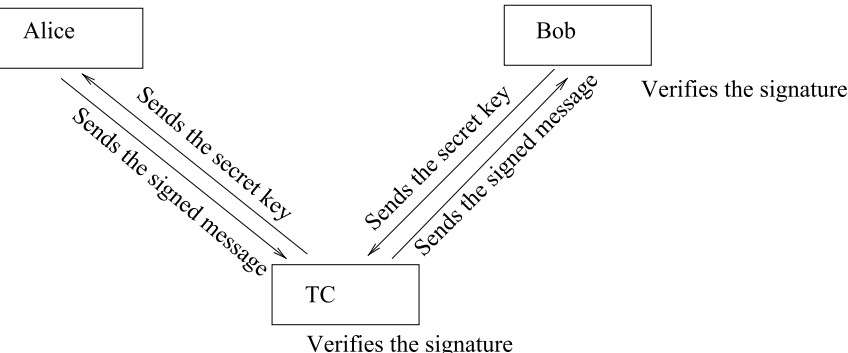

**Fig 1. Quantum signature scheme based on amplitude amplification operators.**

**Table 1. Actions that can be taken and shared on a secret key.**

| Qubit | Symbol | Meaning |
|---|---|---|
| 000 | (H,1) | The data will be sent on the first qubit and is hidden. |
| 001 | (H,2) | The data will be sent on the second qubit and is hidden. |
| 010 | (N,1) | The data will be sent on the first qubit and is not hidden. |
| 011 | (N,2) | The data will be sent on the second qubit and is not hidden. |
| 100 | (B,1) | The data will be in the first qubit and will be encoded as Bell state. |
| 101 | (B,2) | The data will be in the second qubit and will be encoded as Bell state. |
| 110,111 | - | That data is meaningless data. |

signed message, each 3 qubits represent an action that will be taken. For example, 000 means the data will be placed on the first qubit and it will be hidden as shown in Table 1.

**3.1.2 Signature phase.** Alice initializes the two quantum channels that will be used in communication to state $|00\rangle$, then Alice signs the message $|S_A\rangle$ by applying operations on the two qubits according to the actions in the secret key received from TC. If the action in the key is to hide or not to hide the data, then Alice adds an auxiliary qubit $|0\rangle$ and applies $D_p U_f$ on the system, the system will be $D_p U_f |00\rangle \otimes |0\rangle$ where $U_f$ will be chosen according to three criteria: the data to be sent is 0 or 1, the data is on the first or second qubit, and whether the quantum state represents the data will be hidden or not. If the action of the key to encode the qubit in Bell basis, then Alice applies Hadamard gate on the first qubit then CNOT gate on the two qubits as shown in Fig 2.

The first qubit is passed through the Hadamard gate and then both qubits are entangled by a CNOT gate. For example, if the input to the system is $|0\rangle \otimes |0\rangle$, then the Hadamard gate changes the state to

$$(H \otimes I)|00\rangle = \frac{1}{\sqrt{2}}(|0\rangle + |1\rangle) \otimes |0\rangle = \frac{1}{\sqrt{2}}(|00\rangle + |10\rangle). \tag{7}$$

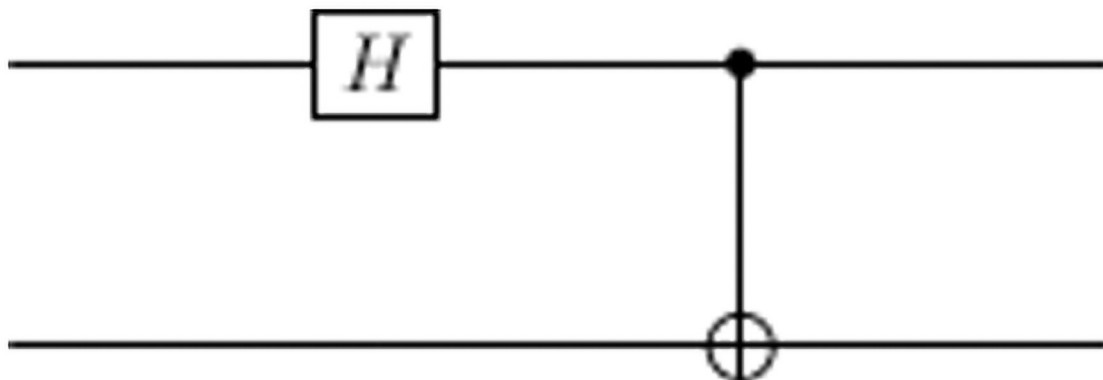

**Fig 2. Circuit to encode qubit to Bell state.**

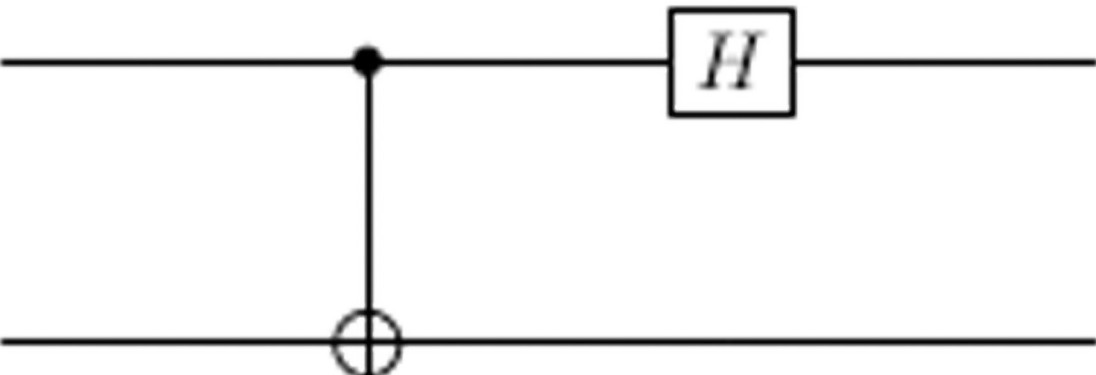

**Fig 3. Circuit to decode Bell states to qubit.**

After applying the CNOT gate, the state becomes $\frac{1}{\sqrt{2}}(|00\rangle + |11\rangle)$. The four possible initializations $\{|00\rangle, |01\rangle, |10\rangle, |11\rangle\}$ will produce the four Bell states as follows,

$$|00\rangle \rightarrow \frac{1}{\sqrt{2}}(|00\rangle + |11\rangle), \qquad\qquad |01\rangle \rightarrow \frac{1}{\sqrt{2}}(|01\rangle + |10\rangle),$$
$$|10\rangle \rightarrow \frac{1}{\sqrt{2}}(|00\rangle - |11\rangle), \qquad\qquad |11\rangle \rightarrow \frac{1}{\sqrt{2}}(|01\rangle - |10\rangle). \tag{8}$$

After signing the message, Alice will send the signed message $|S_A\rangle$ to TC.

**3.1.3 Verification phase.**   TC decrypts the signed message by applying an action on the received signed message according to the key that TC sends to Alice. If the key is 000 or 001, then TC applies diffusion operator $G$ to unhide the states. If the key is 010 or 011, then TC applies measurment $M$ and if the key is 100 or 101, then TC applies CNOT gate on the two qubits then Hadamard gate on the first qubit to decode the qubit $B$ as shown in Fig 3.

TC compares the result of applying the key on the signed message $|S_A\rangle$ with the public message, if they are identical, then TC confirms that Alice is a legitimate signer.

For example, Assume Alice wants to send 1, and the shared key is 001, then the system that will be sent $|\psi\rangle = \frac{1}{\sqrt{2}}(|00\rangle + |10\rangle)$, and TC applies $G$ to decrypt the signed qubit, then measures the second qubit. The system will be as shown in Fig 4.

First the system is initialized to $|000\rangle$, where the third qubit is extra qubit to mark the required states, then Alice prepares a superposition, the system will be $|\psi\rangle = \frac{1}{2}(|00\rangle + |01\rangle + |10\rangle + |11\rangle) \otimes |0\rangle$, then after applying $U_f$ the system will be $|\psi\rangle = \frac{1}{2}(|00\rangle + |10\rangle) \otimes |1\rangle + \frac{1}{2}(|01\rangle + |11\rangle) \otimes |0\rangle$, then after applying $D_p$ operator, the system will be $|\psi\rangle = \frac{1}{\sqrt{2}}(|00\rangle + |10\rangle) \otimes |1\rangle$. Finally, Alice measures the third qubit and sends the signature $|S_A\rangle$ to TC, the system will be $|S_A\rangle = \frac{1}{\sqrt{2}}(|00\rangle + |10\rangle)$, where the third qubit can be omitted.

After TC receives $|S_A\rangle$, TC applies $G$ then measures the second qubit to get the data. As explained in section 2, if TC applies $G$ on the system, then the system will be $|\psi\rangle = \frac{1}{\sqrt{2}}(|01\rangle + |11\rangle)$. After applying measurement on the second qubit, the data is 1. TC compares the result with the public message, they are identical, then TC confirms that Alice is a legitimate signer.

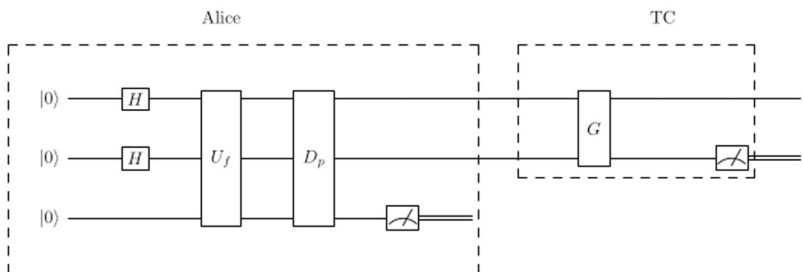

**Fig 4. A quantum circuit to send binary-1 on the second qubit where the data is hidden.**

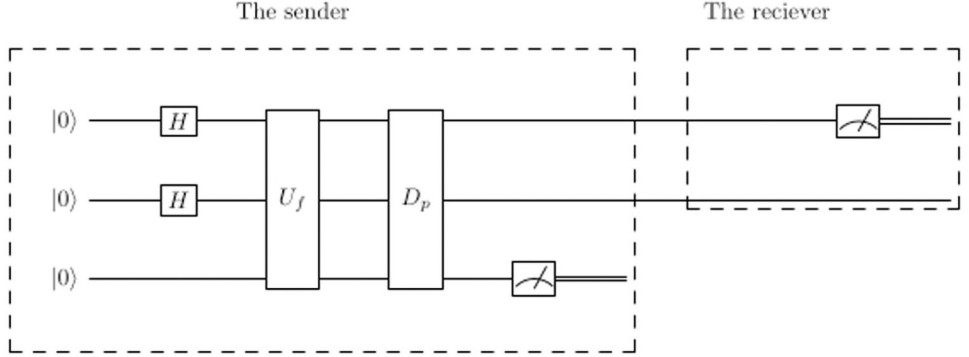

**Fig 5. A quantum circuit to send the data on the first qubit where the data is not hidden.**

Fig 5 shows the system if the data will be sent on the first qubit and is not hidden. Fig 6 shows the circuit if the data is 0 and will send on the first qubit and the data will be encoded as Bell state, where $|x\rangle$ can be either $|0\rangle$ or $|1\rangle$.

## 3.2 Phases between TC and Bob

After confirming that the received message was legitimately signed by Alice, TC and Bob repeat the same phases as above. By this process, Bob can ensure that the message received from Alice is signed by her.

**3.2.1 Preparation phase.** TC and Bob share a secret key $K_{BT}$ using QKD protocol. The key is with size $3S$ qubits with the same form TC sent it to Alice.

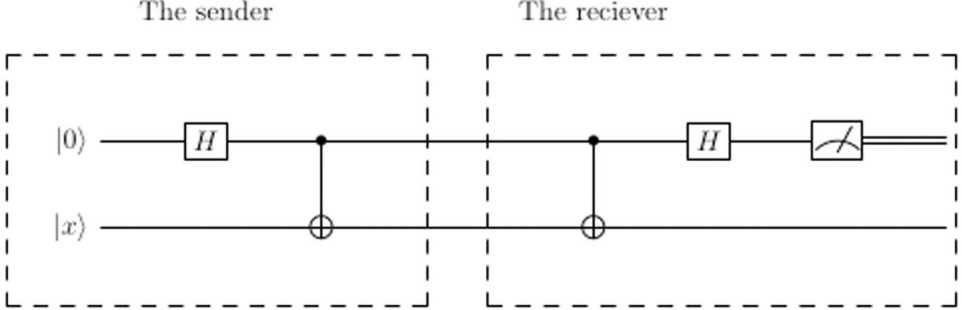

**Fig 6. A quantum circuit to send binary-0 on the first qubit where the data is encoded as Bell state.**

**3.2.2 Signature phase.** TC initializes the two quantum channels that will be used in communication with Bob to state $|00\rangle$. TC signs the message $|S_T\rangle$ by applying operations on the two qubits according to the action in the secret key received from Bob. TC prepares the system $D_p U_f |00\rangle \otimes |0\rangle$ and chooses $U_f$ according to whether or not the data will be hidden. and if the action in the key is to encode the data to Bell basis, then TC applies Hadamard gate on the first qubit then CNOT gate on the two qubits to encode the message in Bell basis.

After signing the message, TC sends the signed message $|S_T\rangle$ to Bob.

**3.2.3 Verification phase.** Bob applies an action on the received signed message according to the key that has been shared with TC. If the key is 000 or 001, then Bob applies diffusion operator $G$ to unhide the states. If the key is 010 or 011, then Bob applies measurement $M$ and if the key is 100 or 101, then Bob applies CNOT gate on the two qubits then Hadamard gate on the first qubit to decode the message. Bob compares the result after applying keys on $|S_T\rangle$ with the public message. If they are identical, then Bob confirms that TC is a legitimate signer. After the above steps, Bob can verify that Alice signed the message legitimately.

# 4 Security analysis

Quantum signature scheme must satisfy the conditions: message authentication, message integrity and non-repudiation [10, 11]. Message authentication is to confirm that the received message was made and signed by the sender. Message integrity means that the legitimately signed message must be protected from forgery by an attacker. If the quantum signature scheme is not satisfying the message authentication or the message integrity, then Bob cannot confirm the signature. Non-repudiation is that Alice cannot repudiate she signed the message and that Bob cannot repudiate that he received the signed message.

## 4.1 Security of message

If Eve can intercept the signed message while transferring qubits, then she neither knows the action applied by the sender nor the position of the signed message. Eve decides randomly to either apply $G$ then measure (action denoted by $GM$), or to directly apply measurement (action denoted by $M$), or to decode Bell state (action denoted by $B$). The receiver randomly chooses the key that represents the action the sender will apply to sign the message, whether to hide data, not hide data or encode qubit in Bell state and sends the action to the sender in the key. Eve does not know the actions taken to sign the message, so Eve decides randomly either to apply measurement assuming that data is not hidden, to apply $G$ assuming data is hidden, or to apply CNOT then Hadamard gates assuming data is encoded in Bell state. After the interception, Eve resends a superposition to the receiver, Eve chooses to resend the measured data without modification, action denoted by $SM$ (Send measured), or to prepare a random superposition and send it, action denoted by $PS$ (Prepare then send). If Eve decides to prepare a superposition, then Eve has to decide whether the data is 0 or 1, the data is on the first or the second qubit, and the action that she will apply on the qubit.

Fig 7 shows the tree of actions taken by sender to sign the message, actions that can be taken by Eve to intercept the message and the action that the receiver applies on the message according to the key that was received from the sender.

To understand the tree of actions, first a sender applies action according to the shared key to sign the message with probability $\frac{1}{3}$, then Eve chooses randomly the action to decode the signature, there are three possibilities, each with probability $\frac{1}{3}$, then Eve decides randomly either resend a measured data without modification with probability $\frac{1}{2}$, or prepare superposition with probability $\frac{1}{2}$. When Eve decides to send prepared superposition, Eve prepares the correct superposition with probability $\frac{1}{6}$. Finally, the receiver chooses the action according to the key

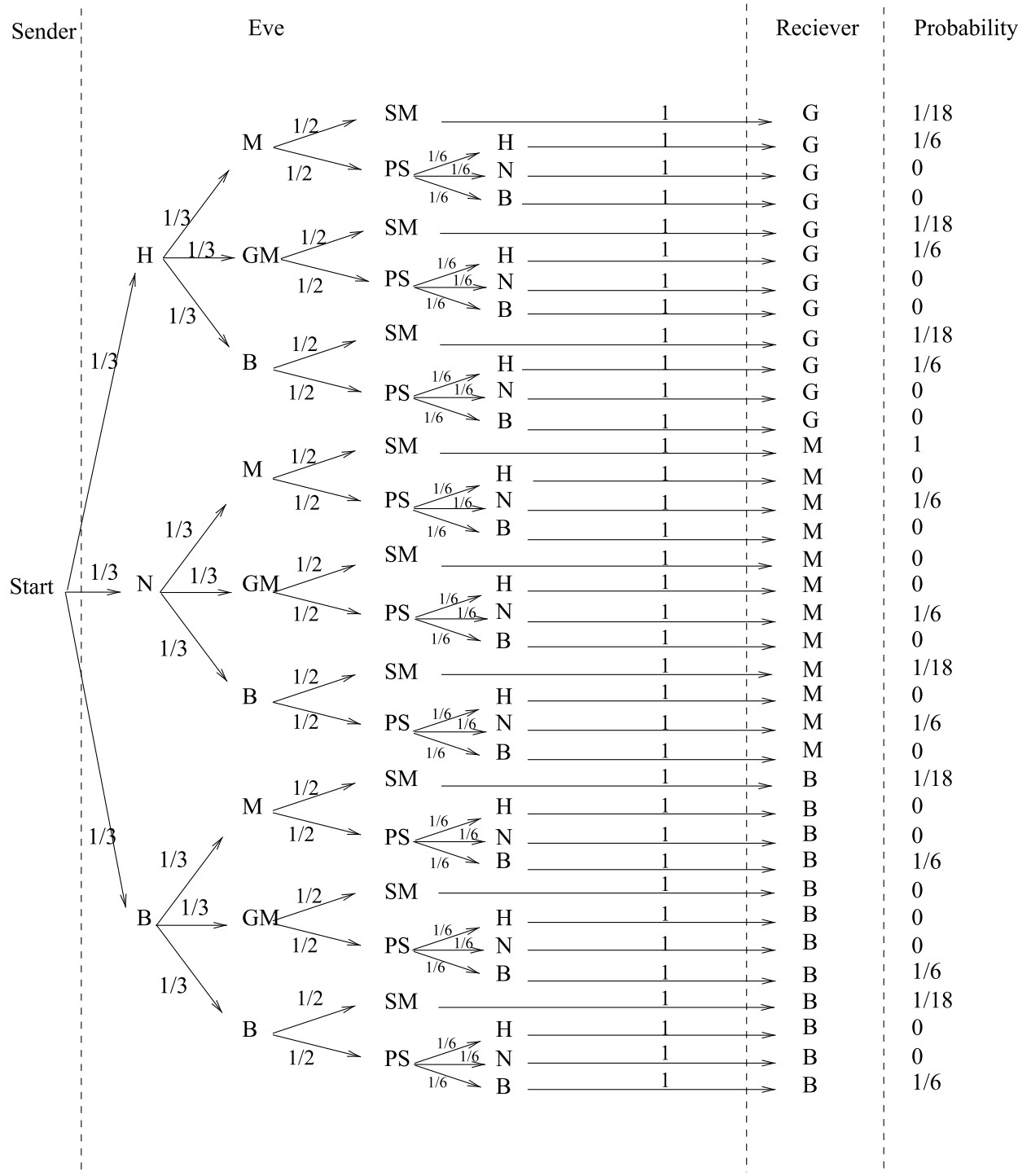

**Fig 7. Tree of actions in the presence of eavesdropping.**

shared between the sender and the receiver. The final probability to get the correct message depends on the actions of Eve. For example, if the key is to hide 1 the first qubit, the signed message is $|S_A\rangle = \frac{1}{\sqrt{2}}(|00\rangle + |01\rangle)$, then the correct action to get the correct data is *GM*. Assume Eve applies *B* action, the system will be $|0\rangle \otimes |0\rangle$, then Eve decides to prepare a random

superposition, where data is encoded in Bell states and send it, assume the system is $|\psi\rangle = \frac{1}{\sqrt{2}}(|00\rangle + |11\rangle)$. When the receiver applies $GM$, the system will be $|\psi\rangle = \frac{1}{\sqrt{2}}(|01\rangle + |10\rangle)$, the data will not be the correct message and Eve can be detected.

## 4.2 The impossibility of forgery

When TC detects the presence of eavesdropping, TC restarts the protocol. Fig 7 shows that the probability to detect eavesdropping by hiding the message will be 98.61%, higher than 93.67% when Alice decides not to hide the message and 98.92% when Alice decides to encode the message in Bell basis. The probability that TC or Bob can't detect the presence of eavesdropping will be 8.796%.

## 4.3 Security against repudiation

Non-repudiation is an important aspect of the digital signature scheme. The signer cannot repudiate that she signed the message, and the receiver cannot repudiate that he received the message. In the proposed scheme, TC confirms the signature, so TC provides non-repudiation [9, 10].

## 4.4 Security against eavesdropping

The efficiency of the proposed scheme can be compared with the existing schemes [9, 11, 13] in terms of the required quantum sources, quantum memory, operations and the number of communications as shown in Table 2, where the proposed quantum signature scheme is more efficient than the other schemes in terms of eavesdropping detection. The source that is used to share the secret key is neglected in the comparison because it changes according to the used QKD protocol. The aim of using QKD in the proposed schemes and our scheme is to share a secret key without determining any protocol. Table 2 shows that the proposed scheme is similar to the scheme in [13] in terms of the quantum source, operations and the number of communications.

In [9, 11], the proposed schemes need quantum memory to transfer the required information and measurements to other parties and need a long-lived memory to save received data and compare it later. Bob and the arbitrator need a long-lived memory to exchange data between them and compare between received data and verify the signature. In [13], a public message is announced on a quantum public channel and the scheme needs a long-lived memory to use the public message to verify the signature to get the required information that verifies the signature. In our scheme, the message that Alice wants to send it to Bob is announced on a classical channel, and that message will be used to verify the signature. Before parties start phases, they share the secret key that will be used to sign the message and verify it, i.e, TC does

**Table 2. Comparison of the main operations in the quantum signature schemes.**

|  | AQS [9] | AQS [11] | QS [13] | Our scheme |
|---|---|---|---|---|
| Quantum source | $S$ Bell states | $S$ Bell states | $2S$ single qubits | $2S$ single qubits |
| Long-lived quantum memory | Yes | Yes | Yes | No |
| Joint MS [18] | $S$ times | $S$ times | 0 | 0 |
| Bell MS [19] | $S$ times | $S$ times | 0 | 0 |
| von Neumann MS | $S$ times | 0 | $2S$ times | $2S$ times |
| A=>B | $4S$ times | $4S$ times | 0 | 0 |
| A=>T=>B | $12S + 1$ times | $10S + 1$ times | $2S$ times | $2S$ times |
| Eavesdropping detection | - | - | 75% | 91.23% |

not need to share the secret key with Bob until TC verifies the signature that Alice sends it. which means the used memory will be used for a short time then another data will be transferred on it.

To compare the security of the proposed scheme with the scheme that uses amplitude amplification operators [13], the proposed scheme in [13] depends on sending a message with decoy qubits with different size in each time where it was shown in [13] that the probability of detecting Eve is 75%, while the proposed scheme depends on using two quantum channels, where one of the qubits will be used to confuse the eavesdropper, and the probability of detecting Eve using the proposed algorithm is 91.23%.

## 5 Discussion

In [12], a proposed method was proposed to secure a quantum communication by hiding specific quantum states in a superposition using a single iteration of partial diffusion operator to hide the quantum states and the diffusion operator to unhide the hidden quantum states. This method uses two quantum channels, where the data can be sent using the first or the second qubit, where the sender prepares and sends the key to the receiver.

In this paper, the proposed quantum signature scheme uses the method shown in [12] as a subroutine to propose a new quantum signature scheme. The proposed scheme consists of three phases: preparation phase, signature phase and verification phase. In the signature phase, if the data to be sent is in the hidden form, then the partial diffusion operator is used by the sender, and the receiver uses the diffusion operator to unhide data and verifies the message, where the key is prepared by the receiver and sent to the sender to use in the preparation phase in contrary to the method shown in [12].

In addition, the proposed quantum signature scheme uses the Bell states as an alternative to the hiding scheme in the encoding of the data in the signature phase, which increases the possibilities for the receiver to choose among six actions and this increased the security of the transmission by confusing the eavesdropper among more choices that that shown in [12]. Using the Bell states increases the probability to detect eavesdropping by hiding the message from 95.3125% in [12] to 98.61% using the proposed method, and the probability to detect eavesdropping by not hiding the message from 85.9375% in [12] to 93.67% using the proposed method. The probability that the receiver cannot detect the presence of eavesdropping is decreased from 18.75% in [12] to 8.796% using the proposed method.

In [12], the sender selects the action applied to the message then sends the key that represents the action to the receiver where the size of action representation is 2-bits. In the proposed quantum signature scheme, the receiver prepares and sends the key to the sender using 3-bits to encode the action due to the addition of the Bell states.

Using the method in [12] as a subroutine in the proposed quantum signature scheme allows the proposed quantum signature not to require a long-lived memory compared to the quantum signature schemes shown in [9, 11, 13] whether to share the key or to verify the message, where the probability of detecting the eavesdropping increases from 75% in [12] to 91.23% using the proposed scheme.

## 6 Conclusion

Digital signature is an important component to be able to establish a secure and trusty communication between two parties. In this paper, a quantum signature scheme is proposed. The proposed scheme uses 2-qubit quantum system to encode every bit in the digital signature. To confuse the eavesdropper, the sender decides to hide the data, not to hide the data or to encode the data using Bell states based on a pre-shared key with the receiver. The process of hiding

and unhiding the quantum states has been achieved using amplitude amplification operators, namely the partial diffusion operator to hide the quantum states and the diffusion operator to unhide the quantum states.

The proposed scheme consists of three phases between the sender and TC, and between TC and the receiver, the preparation phase, the signature phase and the verification phase. During the preparation phase, a key is exchanged between the sender and the receiver using QKD. In the signature phase, the signature is encoded using a 2-qubit system based on the pre-shared key. The verification phase is used to make sure that the sender signed the message legitimately.

The analysis of the proposed algorithm shows that using the Bell states together with the hiding/unhiding technique will enhance the impossibility of forgery so that the presence of eavesdropping can be detected with high probability.

## Acknowledgments

The authors would like to gratefully thank Prof. Youssri Hussien (Alexandria University) for his valuable comments and support on an early stage of this work.

## Author Contributions

**Conceptualization:** Basma Elias, Ahmed Younes.

**Data curation:** Basma Elias.

**Formal analysis:** Basma Elias.

**Methodology:** Ahmed Younes.

**Project administration:** Ahmed Younes.

**Resources:** Ahmed Younes.

**Supervision:** Ahmed Younes.

**Validation:** Basma Elias, Ahmed Younes.

**Visualization:** Basma Elias.

**Writing – original draft:** Basma Elias.

**Writing – review & editing:** Ahmed Younes.

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
