## [Decision Letter · Decision Letter 0]

3 Feb 2021

PONE-D-20-39685

Enhanced quantum signature scheme using quantum amplitude amplification operators

PLOS ONE

Dear Dr. Elias,

Thank you for submitting your manuscript to PLOS ONE. After careful consideration, we feel that it has merit but does not fully meet PLOS ONE’s publication criteria as it currently stands. Therefore, we invite you to submit a revised version of the manuscript that addresses the points raised during the review process.

We look forward to receiving your revised manuscript.

Kind regards,

M. Usman Ashraf, Ph.D

Academic Editor

PLOS ONE

Journal Requirements:

2.We note that the grant information you provided in the ‘Funding Information’ and ‘Financial Disclosure’ sections do not match.

Reviewers' comments:

Reviewer's Responses to Questions

**Comments to the Author**

1. Is the manuscript technically sound, and do the data support the conclusions?

Reviewer #1: Partly

Reviewer #2: Partly

2. Has the statistical analysis been performed appropriately and rigorously? 

Reviewer #1: No

Reviewer #2: N/A

3. Have the authors made all data underlying the findings in their manuscript fully available?

Reviewer #1: No

Reviewer #2: Yes

4. Is the manuscript presented in an intelligible fashion and written in standard English?

Reviewer #1: Yes

Reviewer #2: Yes

5. Review Comments to the Author

Reviewer #1: The paper describes an extension towards the quantum signature scheme based on Grover's quantum search algorithm related diffusion / partial diffusion / amplitude amplification operators along the lines of the publication of A. Younes "Enhancing the security of quantum communication by hiding the message in a superposition" from 2011 (https://doi.org/10.1016/j.ins.2010.09.017). The current submission however does not offer a sufficient conceptual advancement from the mentioned paper to justify its publication in PLOS ONE. The main concept of the paper has been described previously in its major part, and thus both the core of the proposed QDS along with its security analysis have been to a high extent repeated after the previous result largely unchanged on a conceptual level.

The paper does not provide sufficient disclosement of the fact that in terms of the protocol's conceptual layer it is mainly repeating the idea of a previous paper of one of the co-authors from 2011. Hence in its current formulation the proposed protocol lacks the novelty justifying its PLOS ONE publication. If the paper is to be reconsidered for the PLOS ONE publication, a significant (major) revision would be required, first in-detail addressing the issue of basing the proposal on the 2011 paper (including indications of which parts of the protocol are essentially repeated) and secondly clearly explaining the conceptual progress from this previous paper that could justify significant novelty required for the publication. Furthermore the security analysis of the protocol in its current formulation is not very convincing and should be extended to justify the claims made on the security of the protocol. The soundness of the concept should also be elaborated on. Furthermore issue of quantum memory requirement for the proposed QDS should be discussed in view of the recent progress with memory-less QDS.

Reviewer #2: The authors investigate a quantum signature scheme via a trusted center (TC),

based on "scrambling" information using Grover-like inversion about the mean

operators.

The signature scheme consists of 3 phases: preparation, signing, and

verifying. During the preparation phase, Alice and Bob establish a shared

secret key via a QKD scheme run through the TC; Alice also sends the message

to Bob via a public channel. In the signing phase, Alice chooses one of three

possible encoding methods to "scramble" the qubits in her message: hiding,

non-hiding, and Bell-like encoding, then apply the encoding to each of her

qubits. Finally, the verification scheme consist of Bob comparing the message

received via the public channel with the one obtained via decoding, and

agreeing whether they are the same or not.

The authors then show that their scheme is non-forgeable, provides

non-repudiation, and is secure.

Finally, the authors compare their scheme with previous ones in the

literature, in terms of used resources (qubits) and eavesdrop detection

efficiency. They show a slight improvement in the latter (from 75% to ~91%,

compared to Ref. 13).

In my opinion, the paper is a bit hard to read and lacks sufficient rigour. In

particular, the following are unclear:

1. Line 78, there should be a 3rd qubit there in state |1>

2. I assume the message to be signed is a S-qubit quantum state (if yes, can

this be arbitrary or only a product state?). Can the authors clarify that.

E.g., on line 100, "Alice sends the message to the receiver on a public

channel" -> "... public quantum channel"

3. On line 111, what do the authors mean by "QKD protocol with size 3S

qubits"? Is 3S the length of the established key? In that case, the QKD

requires significantly more than 3S qubits to distill a 3S-length key.

4. Regarding the QKD, it is not clear from the text that it has to go through

the TC (as shown in the Fig. 1). Can Alice and Bob perform the QKD without the

TC?

5. Sign is wrong on line 87, the minus should appear on the other 3 states

6. During the verification scheme, the authors mention that the public message

(I assume string of received qubits) is "compared" against the decoded one

received via the channel. This implies the receiver must have access to a

long-lived quantum memory, which should be included in the resource analysis

of Table 2.

7. Can the authors comment more on the role of the TC and the attack model?

What exactly can Eve intercept/modify?

8. The encoding procedure during the signing phase is not very clear. I urge

the authors to write the encoding as a set of 3 quantum circuits, one for each

case, which will make the procedure more clear for the reader.

In light of the above comments, and since the current paper level of

innovation is a bit less than PLOS ONE's standards, I recommend a soft

rejection, unless the authors significantly re-write it, address all the

criticisms above, and clearly emphasize the novel approaches in comparison

with the known literature.

6. PLOS authors have the option to publish the peer review history of their article (what does this mean?). If published, this will include your full peer review and any attached files.

Reviewer #1: No

Reviewer #2: No

---

## [Author Response · Author response to Decision Letter 0]

21 Mar 2021

Reviewer #1: The paper describes an extension towards the quantum signature scheme based on Grover's quantum search algorithm related diffusion / partial diffusion / amplitude amplification operators along the lines of the publication of A. Younes "Enhancing the security of quantum communication by hiding the message in a superposition" from 2011 (https://doi.org/10.1016/j.ins.2010.09.017). The current submission however does not offer a sufficient conceptual advancement from the mentioned paper to justify its publication in PLOS ONE. The main concept of the paper has been described previously in its major part, and thus both the core of the proposed QDS along with its security analysis have been to a high extent repeated after the previous result largely unchanged on a conceptual level.

The paper does not provide sufficient disclosement of the fact that in terms of the protocol's conceptual layer it is mainly repeating the idea of a previous paper of one of the co-authors from 2011. Hence in its current formulation the proposed protocol lacks the novelty justifying its PLOS ONE publication. If the paper is to be reconsidered for the PLOS ONE publication, a significant (major) revision would be required, first in-detail addressing the issue of basing the proposal on the 2011 paper (including indications of which parts of the protocol are essentially repeated) and secondly clearly explaining the conceptual progress from this previous paper that could justify significant novelty required for the publication. 

*Author response: Thank you very much for the comment. The paper has been reorganized by adding section 2 to separate the work done in ( https://doi.org/10.1016/j.ins.2010.09.017) from the contribution of the paper. 

Furthermore the security analysis of the protocol in its current formulation is not very convincing and should be extended to justify the claims made on the security of the protocol. The soundness of the concept should also be elaborated on.

*Author response: Thank you very much for the comment. The security analysis of the protocol has been discussed and explained in page 11, lines from 222 to 236.

 Furthermore issue of quantum memory requirement for the proposed QDS should be discussed in view of the recent progress with memory-less QDS.

*Author response: Thank you very much for the comment. The issue of quantum memory requirement is discussed and explained in page 12 and 13, lines from 259 to 271.

Reviewer #2: 

1. Line 78, there should be a 3rd qubit there in state |1>

Author response: Thank you very much for the comment.

2. I assume the message to be signed is a S-qubit quantum state (if yes, can this be arbitrary or only a product state?). Can the authors clarify that.

E.g., on line 100, "Alice sends the message to the receiver on a public channel" -> "... public quantum channel"

Author response: Thank you very much for your valuable comment that enables us to enhance and clarify the paper. The announcement of the message is explained in page 6, lines 115 and 116.

3. On line 111, what do the authors mean by "QKD protocol with size 3S qubits"? Is 3S the length of the established key? In that case, the QKD requires significantly more than 3S qubits to distill a 3S-length key.

Author response: Thank you very much for your valuable comment that enables us to clarify the paper. The requirement of the QKD protocol is discussed and explained in page 6, lines from 117 to 120.

4. Regarding the QKD, it is not clear from the text that it has to go through the TC (as shown in the Fig. 1). Can Alice and Bob perform the QKD without the TC?

Author response: Thank you very much for the comment. In the scheme, TC sends a secret key to Alice to sign the message (page 6, line 117) then TC verifies the signature using the shared key, then TC sends the signed message to Bob based on a shared secret key that Bob sends to TC (page 9, line 173). If Alice and Bob perform the QKD without TC, TC cannot verify the signed message that Alice sends because TC does not have a secret key and cannot sign the message again to send it to Bob. 

5. Sign is wrong on line 87, the minus should appear on the other 3 states

Author response: Thank you very much for the comment. We would like to inform you that the equation is correct, and the minus sign is on state |00> only because Grover’s operator is applied on state |00> (no perfect superposition and no oracle for marking).

6. During the verification scheme, the authors mention that the public message

(I assume string of received qubits) is "compared" against the decoded one

received via the channel. This implies the receiver must have access to a

long-lived quantum memory, which should be included in the resource analysis

of Table 2.

Author response: Thank you very much for your valuable comment that enables us to clarify the paper. The announcement of the message is explained in page 6, line 115 and 116. The issue of quantum memory requirement is discussed and explained in page 12 and 13, lines from 259 to 271. The need of the long-lived quantum memory is included in the resource analysis

of Table 2.

7. Can the authors comment more on the role of the TC and the attack model?

What exactly can Eve intercept/modify?

Author response: Thank you very much for the comment. The security analysis of the protocol has been discussed and explained in page 11, lines from 222 to 236.

8. The encoding procedure during the signing phase is not very clear. I urge the authors to write the encoding as a set of 3 quantum circuits, one for each case, which will make the procedure more clear for the reader.

Author response: Thank you very much for your valuable comment that enables us to clarify the paper. The cases of the signing phase are explained and discussed in page 8, lines from 150 to 167.

---

## [Decision Letter · Decision Letter 1]

15 Jun 2021

PONE-D-20-39685R1

Enhanced quantum signature scheme using quantum amplitude amplification operators

PLOS ONE

Dear Dr. Elias,

Thank you for submitting your manuscript to PLOS ONE. After careful consideration, we feel that it has merit but does not fully meet PLOS ONE’s publication criteria as it currently stands. Therefore, we invite you to submit a revised version of the manuscript that addresses the points raised during the review process.

We look forward to receiving your revised manuscript.

Kind regards,

M. Usman Ashraf, Ph.D

Academic Editor

PLOS ONE

Reviewers' comments:

Reviewer's Responses to Questions

**Comments to the Author**

1. If the authors have adequately addressed your comments raised in a previous round of review and you feel that this manuscript is now acceptable for publication, you may indicate that here to bypass the “Comments to the Author” section, enter your conflict of interest statement in the “Confidential to Editor” section, and submit your "Accept" recommendation.

Reviewer #2: (No Response)

2. Is the manuscript technically sound, and do the data support the conclusions?

Reviewer #2: Partly

3. Has the statistical analysis been performed appropriately and rigorously? 

Reviewer #2: N/A

4. Have the authors made all data underlying the findings in their manuscript fully available?

Reviewer #2: Yes

5. Is the manuscript presented in an intelligible fashion and written in standard English?

Reviewer #2: Yes

6. Review Comments to the Author

Reviewer #2: I appreciate that the authors addressed some of my concerns, and clarify some

of the material in the paper.

My only worry is that the current paper is not innovative enough in comparison

with Ref. 12. I appreciate that the authors introduced Sec. 2, in which they

succinctly describe Ref. 12, however I find the progress made in the current

manuscript to be relatively incremental compared to Ref. 12, and therefore

does not meet the PONE standards of novelty.

For this reasons, unfortunately I cannot recommend the publication.

7. PLOS authors have the option to publish the peer review history of their article (what does this mean?). If published, this will include your full peer review and any attached files.

Reviewer #2: No

---

## [Author Response · Author response to Decision Letter 1]

1 Jul 2021

Reviewer #2: I appreciate that the authors addressed some of my concerns, and clarify some

of the material in the paper.

My only worry is that the current paper is not innovative enough in comparison with Ref. 12. I appreciate that the authors introduced Sec. 2, in which they succinctly describe Ref. 12, however I find the progress made in the current manuscript to be relatively incremental compared to Ref. 12, and therefore does not meet the PONE standards of novelty.

*Author response: Thank you very much for the comment. The paper has been reorganized by adding section 5 to discuss the innovation done in our work compared with Ref. 12.

---

## [Editor Report · Decision Letter 2]

20 Sep 2021

Enhanced quantum signature scheme using quantum amplitude amplification operators

PONE-D-20-39685R2

Dear Dr. Elias,

We’re pleased to inform you that your manuscript has been judged scientifically suitable for publication and will be formally accepted for publication once it meets all outstanding technical requirements.

Kind regards,

M. Usman Ashraf, Ph.D

Academic Editor

PLOS ONE

Additional Editor Comments (optional):

The basic idea of the research is although very innovative. After the 2nd revision, I can see improvement in the paper. However, my decision is to accept the paper.   

Reviewers' comments:

Authors have addressed all the comments very carefully. I recommend to accept the paper in its present format.

---

## [Editor Report · Acceptance letter]

27 Sep 2021

PONE-D-20-39685R2 

Enhanced quantum signature scheme using quantum amplitude amplification operators 

Dear Dr. Elias:

I'm pleased to inform you that your manuscript has been deemed suitable for publication in PLOS ONE. Congratulations! Your manuscript is now with our production department. 

Kind regards, 

on behalf of

Dr. M. Usman Ashraf 

Academic Editor

PLOS ONE